# T2-PILOT: Optimized Trajectories for $T_2$ Mapping Acceleration

**Naama Gavrielov**[1] (iD)                                    NAAMAGAV@CAMPUS.TECHNION.AC.IL

[1] *Faculty of Biomedical Engineering, Technion – Israel Institute of Technology, Haifa, Israel*

**Tamir Shor**[2] (iD)                                    TAMIR.SHOR@CAMPUS.TECHNION.AC.IL

[2] *The Taub Faculty of Computer Science, Technion – Israel Institute of Technology, Haifa, Israel*

**Alex Bronstein**[2,3]                                    BRON@CS.TECHNION.AC.IL

[3] *Institute of Science and Technology Austria*

**Moti Freiman**[1] (iD)                                    MOTI.FREIMAN@TECHNION.AC.IL

## Abstract

Cardiac MRI $T_2$ mapping is essential for diagnosing myocardial pathologies, but prolonged acquisitions often extend beyond the diastolic rest phase, leading to motion artifacts and reduced reliability. While deep learning accelerates imaging via k-space undersampling, existing learned trajectories optimize for image reconstruction and neglect the underlying $T_2$ relaxation physics. We propose T2-PILOT, which jointly optimizes non-Cartesian k-space trajectories and $T_2$ map estimation by enforcing the $T_2$ decay model during training, with additional subject-specific test-time fine-tuning. On the CMRxRecon dataset, T2-PILOT outperforms both fixed and unconstrained reconstruction-guided trajectories. Under high undersampling, it improves image quality and quantitative $T_2$ accuracy while reducing per-beat acquisition time by 54% (32 vs. 70 spokes), yielding a PSNR gain of 0.14 dB (35.06 vs. 34.92) and a $T_2$ map accuracy improvement of 0.67 dB (31.96 vs. 31.29). These results demonstrate that incorporating physics-based constraints into trajectory learning enables more accurate, robust, and clinically reliable accelerated $T_2$ mapping.

**Keywords:** Cardiac MRI, $T_2$ Mapping, Trajectory Optimization and Reconstruction, Physics-Informed Deep-Learning

## 1. Introduction

Cardiac MRI $T_2$ mapping is essential for diagnosing myocardial pathologies (Giri et al., 2009; Ugander et al., 2012). Standard methods (e.g., $T_2$-prepared TrueFISP (Huang et al., 2007)) acquire multiple images at varying echo times to fit an exponential decay model: $M(t) = M_0 \cdot e^{-t/T_2}$. However, these sequences require extended acquisition windows, typically taking ~200 ms per heartbeat for fully sampled data, often forcing data collection outside the quiescent diastolic phase. This prolonged duration makes them highly susceptible to cardiac motion artifacts (Scott et al., 2009; Fenski et al., 2025).

Deep learning and compressed sensing (Lustig et al., 2007) mitigate this via k-space undersampling. Yet, current acceleration methods typically rely on fixed acquisition schemes (Huang et al., 2012; Feng et al., 2011) or optimize learned trajectories purely for image reconstruction (Chaithya et al., 2022; Shor et al., 2024; Yiasemis et al., 2024). Failing to integrate the physical $T_2$ decay model ignores the feasible set of solutions and requires higher-dimensional optimization, yielding sub-optimal results.

To address this, we introduce T2-PILOT, an adaptation of T1-PILOT (Shor et al., 2025), extending it to explicitly model $T_2$ relaxation within the joint optimization of physically feasible, non-Cartesian per-frame k-space trajectories, a reconstruction network, and $T_2$ map estimation. By incorporating the $T_2$ decay model as an explicit optimization constraint, T2-PILOT tightly couples learned acquisition with physics-driven reconstruction.

## 2. Methods

We jointly optimize a non-Cartesian k-space trajectory $\psi$ and reconstruction weights $\zeta$ to recover dynamic $T_2$-weighted cardiac images. To stabilize joint optimization, we utilize a three-stage schedule using fully sampled $T_2$-weighted images $\mathbf{X} = \{x_{t_i}\}_{i=1}^N$ acquired at echo times $\mathbf{T}_E = \{t_i\}_{i=1}^N$: 1) Reconstruction Pre-Training: Let $\mathcal{F}_{K_\psi}$ denote the forward undersampling operator parameterized by trajectory $\psi$. We first optimize a TEAM-PILOT (Shor et al., 2024) network $\mathcal{R}_\zeta$ and trajectory $\psi$ using a standard reconstruction loss $\|\mathbf{X} - \mathcal{R}_\zeta(\mathcal{F}_{K_\psi}(\mathbf{X}))\|_2^2$ to establish physically feasible trajectories. 2) Global Decay Optimization: We append a decay block $\mathcal{D}_\xi$ to extract the parameters of the $T_2$ decay signal ($M_0$ and $T_2$). To explicitly guide the exponential regression, an MLP $\mathcal{M}_\theta$ maps the varying echo times $\mathbf{T}_E$ to learned temporal embeddings that are added to each reconstructed input sample. The full model is optimized end-to-end using the physics-based objective:

$$\min_{\psi,\zeta,\xi,\theta} \sum_{t_i \in \mathbf{T}_E} \left\| x_{t_i} - \left( M_0 \cdot e^{-t_i/T_2} \right) \right\|_2^2$$

where $(M_0, T_2) = \mathcal{D}_\xi(\mathcal{R}_\zeta(\mathcal{F}_{K_\psi}(\mathbf{X})) + \mathcal{M}_\theta(\mathbf{T}_E))$. 3) Subject-Specific Refinement (Optional): At inference, the sampling trajectory $\psi$ remains frozen while the weights $\{\zeta, \xi, \theta\}$ are fine-tuned on the acquired undersampled test sequence.

## 3. Experiments

We evaluated T2-PILOT on the CMRxRecon dataset (Wang et al., 2023), using 3T $T_2$-weighted images (3 echoes at $\mathbf{T}_E = 0, 35, 55$ ms, matrix size $116 \times 384$), split 80/20 for training and validation. Reference $T_2$ maps were generated via least-squares fitting on the fully-sampled data. Training comprised 150 epochs of reconstruction pre-training, 200 epochs of global decay optimization, and 3000 iterations of per-sample fine-tuning.

To isolate the impact of our physics-driven pipeline, we conducted experiments across our three optimization stages: **+Recon** (trajectory optimization guided solely by reconstruction loss), **+Decay** (joint optimization explicitly constrained by the $T_2$ decay objective), and **+Fine-tuning** (test-time subject-specific refinement with frozen trajectories). We evaluated this progression across three k-space schemes: **Fixed Golden-Angle Radial (GAR)** (Zhou et al., 2017), **Learned Radial**, and **Learned Cartesian**, and across three acceleration rates: 16, 32, and 64 shots (k-space lines). Performance was evaluated using myocardium-masked PSNR for image quality, $T_2$ map accuracy (via least-squares fitting), and regional $T_2$ bullseye plots for the 32-shot experiment.

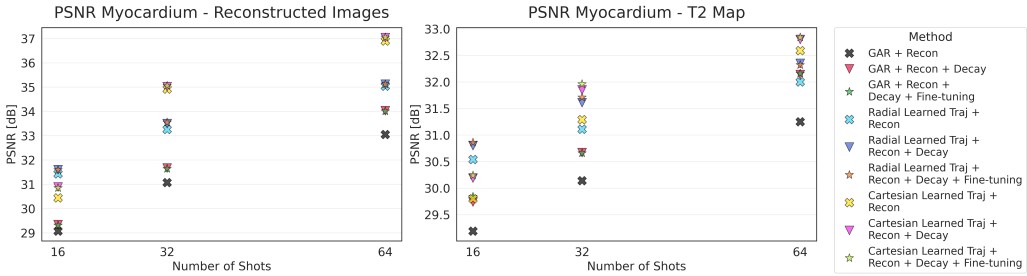

Figure 1: PSNR comparison of reconstructed myocardial images and $T_2$ maps

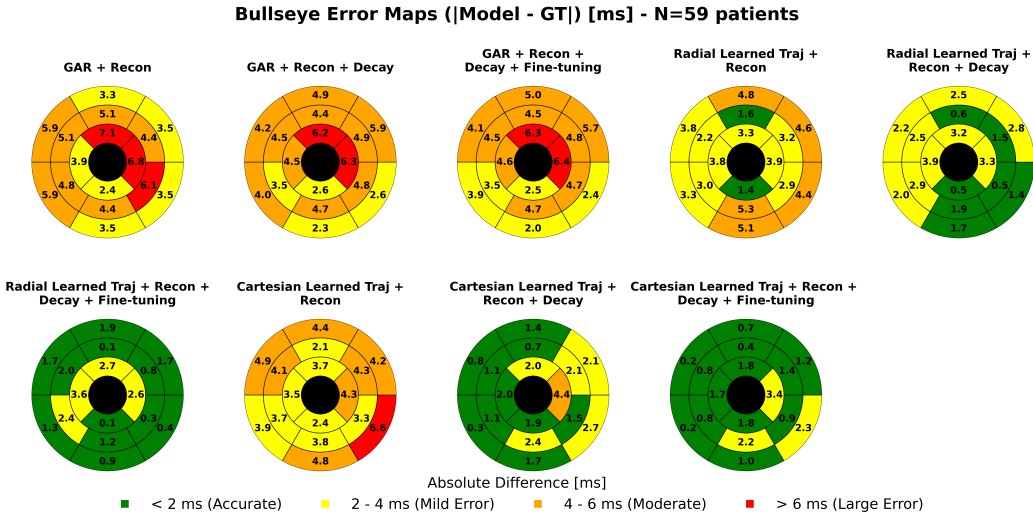

Figure 2: Myocardial $T_2$ bullseye plots comparing absolute reconstruction errors for the 32-shot acceleration experiment.

## 4. Results

Figure 1 and Figure 2 show myocardial PSNR across acceleration rates and 32-shot $T_2$ bullseye plots, respectively, for all methods, with segment values indicating absolute mean $T_2$ error relative to pseudo-ground truth. T2-PILOT outperforms both fixed and unconstrained learned trajectories; under high undersampling, it improves image quality and quantitative $T_2$ accuracy, achieving a myocardial PSNR gain of 0.14 dB (35.06 vs. 34.92) and a $T_2$ map PSNR improvement of 0.67 dB (31.96 vs. 31.29), while reducing per-beat acquisition time by 54% (32 vs. 70 spokes). Overall, learned trajectories surpass the fixed GAR baseline, explicit decay constraints consistently enhance myocardial PSNR and $T_2$ accuracy, and subject-specific fine-tuning provides stable refinement with marginal gains in $T_2$ accuracy without degrading image quality.

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
