# OpenReview forum: "T2-PILOT: Optimized Trajectories for $T_2$ Mapping Acceleration"
_MIDL.io/2026/Short_Papers — MIDL 2026 - Short Papers Poster_

### Official Review · Reviewer_a61h · 2026-05-04
**Physics-informed kspace sampling trajectory learning for T2 mapping**

**Rating:** 3
**Confidence:** 4

**Review:**

Overall, the paper is well-motivated and technically sound. Incorporating the T2 decay model into trajectory optimization is intuitive and aligns with the goal of task-driven MRI acquisition.
Strengths:
- Clear physics-informed formulation
- Joint optimization of acquisition and estimation
- Consistent improvements across settings
- Clinically relevant application

Weaknesses:
- Gains are relatively small (e.g., ~0.1 dB PSNR)
- Novelty is somewhat incremental over prior physics-informed / task-driven methods
- Limited analysis (e.g., ablations, robustness, stronger baselines)
- Some components (e.g., temporal embedding) lack justification
- Test-time fine-tuning adds complexity with marginal benefit

Overall, while the idea is solid, the paper would benefit from stronger empirical evidence and deeper analysis of why the method works.

**Summary:**

This paper proposes T2-PILOT, a method that jointly optimizes k-space trajectories, reconstruction, and T2 estimation by enforcing the exponential decay model. The framework integrates physics constraints into trajectory learning and includes optional test-time refinement. Experiments on CMRxRecon show consistent but small improvements over fixed and unconstrained learned trajectories in PSNR and T2 accuracy. The work highlights the benefit of aligning acquisition design with quantitative imaging objectives.

**Strengths:**

The paper presents a clean and well-motivated integration of physics constraints into trajectory learning for quantitative MRI. The joint optimization framework is technically sound and aligns well with task-driven imaging goals. The method is evaluated across multiple sampling schemes and consistently improves both reconstruction quality and T2 estimation. The problem is clinically relevant, and the approach is conceptually appealing.

**Weaknesses:**

The main limitation is the modest improvement over baselines, which raises questions about practical impact. The novelty is incremental relative to existing physics-informed and task-driven acquisition methods. The paper lacks deeper analysis, including ablations and robustness studies. Some design choices are not well justified, and the benefit of test-time fine-tuning appears limited compared to its added complexity.

**Justification Of Rating:**

The method is reasonable and well-executed, but the contribution is incremental and the improvements are modest. Stronger experimental validation and deeper analysis would be needed to clearly justify acceptance.

---

### Decision · Program_Chairs · 2026-05-08

Accept (Poster)